# Inferring leader-follower behavior from presence data in the marine environment: a case study on reef manta rays

Juan Fernández-Gracia[1][*], Jorge P. Rodríguez[2], Lauren R. Peel[3], Konstantin Klemm[1], Mark G. Meekan[3],and Víctor M. Eguíluz[1]

**1** Instituto de Física Interdisciplinar y Sistemas Complejos IFISC (CSIC-UIB), E07122 Palma de Mallorca, Spain
**2** IMEDEA
**3** Australian Institute of Marine Science, Indian Ocean Marine Research Centre (IOMRC), University of Western Australia (M470), 35 Stirling Highway, Crawley, WA 6009, Australia
* juanf@ifisc.uib-csic.es

April 12, 2022

## Abstract

Social interactions are ubiquitous in groups of animals, including humans. These interactions might be of different nature, for example, competitive, mutualistic, or kinship; and their global structure is naturally studied with the tools of complex network theory. Traditionally, it has been challenging to examine social interactions in the marine environment given the difficulties associated with data collection, however, developments in acoustic telemetry technologies now present a novel way to remotely examine such behaviour. Here, we propose a method to extract leader-follower networks from presence data collected by a single acoustic receiver at a specific location. The method is based on the Kolmogorov-Smirnov distance between the distribution of lag times between the consecutive presence of an individual followed by the presence of another and its conjugate distribution. After characterising the method through controlled generated data, we apply it to detection data collected for acoustically tagged reef manta rays (Mobula alfredi) by a single acoustic receiver positioned at a frequently visited site. First, we show that the presence of reef manta rays in the vicinity of the cleaning station displays several temporal heterogeneities, such as a circadian rhythm, as well as burst-like behavior, where the time between consecutive presences follows a power-law distribution. Second, we infer the leader-follower network of manta rays and characterize individuals in terms of their position on this network relative to their sex and size. We find, in agreement with biological and ecological insights, that (i) female reef mantas follow more males than expected; (ii) male reef manta rays follow fewer females than expected, but with a stronger association to certain individuals; (iii) reef manta rays follow each other with a weaker association than larger individuals do, while the rest of the reported interactions between individuals appear to be random.

## 1 Introduction

Social interactions among animals mediate many processes, such as the transmission of information [1, 2] or diseases [3]; collective behavior [4–6] (flocking [7], social learning [8], predator avoidance [9], cooperative foraging [10]); selection of phenotypes [11]; mating [10]; or the emergence and maintenance of cooperation [10,12]. The structure of these social interactions as a whole is typically studied with the methods from network theory [13], which have

recently and increasingly been applied to animal groups [1–10,14] and represent a promising tool in the field of movement ecology [15].

The first step of studying these interactions involves finding or collecting the data that describe them, before the underlying networks can be extracted in a meaningful and statistically significant way. For humans, the availability of large amount of digital traces (notably, call detail records and social media accounts [16]) and the ease at which they can be accessed, has facilitated the detailed statistical description of human interactions. Tracking animal social systems [17] in a similar manner, however, has remained a challenge, given the difficulties associated with collecting a sufficient amount of data from the network of interest. This is particularly true in the marine environment, where the large and three-dimensional nature of animal movements impose logistical, technological and financial constraints on data collection capacities.

Traditionally, the collection of presence data in the marine environment relied on techniques that were applied within discrete sampling periods (e.g. photo-identification [18] or capture-mark-recapture [19,20]). This temporal limitation restricted the extent to which social interactions between sampled individuals could be considered. Recent advances in the field of acoustic telemetry, however, have since provided a means to overcome these issues, allowing the movement and residency patterns of marine fauna to be monitored continuously, and over long periods of time (up to ten years).

In studies using passive acoustic telemetry, acoustic tags that emit a unique sound signal are externally attached to [21], or surgically implanted into [22], study animals. These tags are subsequently detected by acoustic receivers placed at specific locations within the study area (collectively, an acoustic array), and the timestamp of each detection is logged by the devices from which data can be downloaded at a later date. Acoustic arrays and tagging programs have been established worldwide [23–25], and the frequency of their use is increasing as the affordability of the required equipment improves [23,26]. While the primary motivation of such studies is often to obtain spatial data to inform the development of conservation measures [21,23,27], the potential to use these presence data to examine leader-follower behavior in marine species is yet to be extensively explored.

Leadership behavior has been reported in many animal groups, including insects (e.g. ants) and birds (e.g. migrating storks), and has been found to be of paramount importance in achieving coordination among individuals [28–30]. Should a population lack clear leaders or a hierarchical structure [31], individuals may have preferences on whom they should interact with, subsequently modifying the strength of the social interaction. Additionally, interactions within a social network may not be reciprocal, generating directionality in the flow of information. This allows for the introduction of focal individuals and their leading and following connections, such that the focal individuals' dynamics are coupled with that of its leaders, while that of its followers are coupled with the focal individual's.

Within a typical social network, individuals are represented by nodes, and the interactions recorded between them are represented by connections (or edges). The power of networks to represent social interactions lies within the ability to characterize edges relative to the type of interaction that has occurred. Interactions can be symmetrical, representing exchanges from one peer to the other and vice versa, or directed, originating from a source towards a target. In this sense, leadership behavior can be examined with network tools, too, assuming that the sources are leaders and the targets are followers. However, appropriate methods for detecting these asymmetric relations are currently lacking.

One can find several social network inference methods in the literature that are well suited for acoustic telemetry data [14,32–38]. The majority of these rely on the co-occurrences of individuals, and thus have the problem of the Gambit of the group (GoG [14]). That is, every pair of animals observed in the same group is treated as having interacted with one another,

although this might appear due to coincidence in visitation patterns and not social affiliation. Early analytical methods also required an appropriate temporal window to be defined, within which co-occurrences were considered [36–39]. This restriction, and that of the GoG, was resolved by the introduction of the 'GMMevents' method by Psorakis and collaborators [32, 33]. These methods allowed researchers to investigate the complex social structure of different animal groups, uncovering their relation to individual characteristics, such as sex, size, personality or genetic traits [11, 34, 40, 41].

Here, we aim to develop an analytical method to provide evidence of leader-follower interactions from acoustic telemetry data, alongside estimates for the statistical significance of the observed interactions. We first describe a novel method for extracting leader-follower interactions from established social networks. We then apply this method to real-world, presence data collected for reef manta rays (Mobula alfredi) at a single location using passive acoustic telemetry. Reef manta rays present an ideal study species for work of this nature for a number of reasons: (1) they show repeated and prolonged visitation to key sites on coral reefs known as cleaning stations, where individual mantas socialize, and cleaner fishes remove parasites from their bodies [42]; (2) individuals are long-lived and can be accurately identified from unique pigmentation patterns displayed on their bodies throughout their lifetime [18, 43]; (3) reef manta rays exhibit two distinct sexes and the size of an individual can be used as a proxy for its maturity status, allowing leader-follower behaviors to be considered relative to these biological traits. Finally, we discuss the applicability of this novel analysis procedure to existing acoustic telemetry datasets, and the potential to expand these works to investigate the leader-follower behavior of populations across entire acoustic arrays.

## 2 Materials and methods

All the methods described here have been implemented computationally using Python 3 and are available through the page in Ref. [44].

### 2.1 Leader-follower interactions from presence data

We propose a measure for quantifying leadership in multiple individual time series. In particular, our method is relevant for examining the collective movement patterns of entities, be it active matter, robots, humans or animals, where presence data of individuals has been recorded at a specific location.

The data for each individual consists of an ordered set of timestamps at which the individual is located in the vicinity of that particular location. In order to infer if individual $i$ typically follows individual $j$, we hypothesize that the time delay, or lag time, between the consecutive detection of first $i$ and then $j$ will be longer than the reverse (*i.e.* the consecutive times of detecting $j$ followed by $i$). We therefore extract the lag times for $j$ followed by $i$, $\{t_{ij}\}$, and $i$ followed by $j$, $\{t_{ji}\}$, and their corresponding distributions $p_{ij}(t)$ ($p_{ji}(t)$), such that $p_{ij}(t)dt$ ($p_{ji}(t)dt$) measures the probability that a randomly selected lag time of $i$ after $j$ ($j$ after $i$) lays in the interval $(t, t + dt)$. We then compute the distance between the distributions of lag times as the Kolmogorov-Smirnof distance [45, 46] ($D_{KS}$) between them, where $D_{KS}$ is defined as the maximum distance between the two cumulative distribution functions $P_1$ and $P_2$ as

$$D_{KS}(P_1, P_2) = \max_t(|P_1(t) - P_2(t)|). \tag{1}$$

We use a signed version of $D_{KS}$ to distinguish which of the two cumulative distributions is greater than the other. Since this quantity is no longer a metric, we now call it the Kolmogorov-Smirnof arrow $A_{KS}$.

$$A_{KS}(P_1, P_2) = P_1(\tau) - P_2(\tau), \tag{2}$$

where $\tau = \arg\max_t(|P_1(t) - P_2(t)|)$. For measuring $A_{KS}$, we find the time, $\tau$, that maximizes the distance between the two compared cumulative distributions, and then draw an arrow from the second to the first distribution (first and second refers to their order in the arguments of $A_{KS}$). If the arrow goes upwards, $A_{KS} = D_{KS}$ and otherwise $A_{KS} = -D_{KS}$. For two interevent sequences 1 and 2, $A_{KS}(P_1, P_2) > 0$ implies that times associated with process 1 are shorter than those associated with process 2. Thus, if individual $j$ is following individual $i$, we will expect $A_{KS}(P_{ij}, P_{ji}) > 0$, while $A_{KS}(P_{ij}, P_{ji}) < 0$ reflects the opposite ($i$ following $j$). The strength of the interaction will be given by the absolute value of the arrow. The value of the arrow is related to the excess of probability to have smaller values than $\tau$ of the cumulative distribution that lies above.

## 2.2  Application to a presence toy model

We create synthetic presence data for a pair of individuals (hereafter, Individual A and Individual B) proposing a model of temporal sequences. The first set of times, corresponding to Individual A, is given by a homogeneous Poisson process of rate $\lambda_1$, i.e., Individual A appears randomly at a constant rate. The temporal sequence for Individual B follows from a non-homogeneous Poisson process with rate $\lambda_2(t)$, which depends on time in such a way that it has a constant rate $\lambda^*$ plus an excess rate $\delta$ in the interval $(-\lambda^*, \infty)$ during an amount of time $\Delta t$ immediately after the presence of Individual A is recorded. This is similar to a Hawkes process [47] and to the model for correspondence by Malmgren et al. [48], but in this case the excitations are given by an independent source and are not self-excitations. If $\Delta t = 0$, we just have two independent homogeneous Poisson processes of rates $\lambda_1$ and $\lambda_2(t) = \lambda^*$, wheras for $\Delta t \to \infty$, we have two independent homogeneous Poisson processes of rates $\lambda_1$ and $\lambda_2 = \lambda^* + \delta$. For an intermediate $\Delta t$, Individual B may attempt to actively be present at, or avoid, the location for a period of time $\Delta t$, shortly after Individual A was there for a positive, or negative, value of $\delta$, respectively. If $\delta = 0$, we again have two independent Poisson processes of rates $\lambda_1$ and $\lambda_2 = \lambda^*$. In the following, we choose $\lambda_1 = 1$ without loss of generality. From the time series for Individuals A and B we extract the sets of lag times $\{t_{12}\}$ and $\{t_{21}\}$ and compute the Kolmogorov-Smirnof arrow $A_{KS}(\{t_{12}\}, \{t_{21}\})$ so that if $A_{KS} > 0$, Individual A is following Individual B, and if $A_{KS} < 0$ Individual B is following Individual A. Several realizations of the process were completed, and the distribution of arrows compiled to examine whether the KS arrow is able to capture the aforementioned leader-follower behaviours (see Figures 1 and 2).

For two independent Poisson processes of different rates ($\delta = 0, \lambda_1 \neq \lambda_2$), the distribution of KS arrows are always symmetric, with low values centered around, but not equal to, zero. This reflects the fact that the sequences are uncorrelated, and that it is not possible to discern if one individual is following the other. The absence of values around zero is related to the fact that the KS arrow will only be zero if the compared distributions are exactly equal. This equality will be only occur when the sampling is infinite, which was confirmed by drawing longer and longer time series sequences and observing that the distribution collapsed towards a single delta function at zero (not shown here). Regarding the correlated case, we checked how the distribution of arrows changes as we change the parameters $\delta$ and $\Delta t$, setting $\lambda^* = 1$ (not shown here). If $\delta = 0, \Delta t = 0$ or $\Delta t$ much bigger than $1/\lambda_1$ the distribution is symmetric around 0 with values close to it, reflecting again that there is no evidence of one individual following the other, as expected for independent presence sequences. For $\delta < 0$, the distribution is non-zero only for positive values of the arrow, indicating that the first individual follows the second (although the model is just assuming that the second one is avoiding the first one), while for $\delta > 0$ the distribution is shifted to negative values of the arrow, showing that the second individual is following the first.

Lastly, we examined the behavior of the distribution of KS arrows when the two time series

sequences were reshuffled in such a way that both individuals retained the same number of events, but without correlations, if there were any. For both the correlated and uncorrelated cases, the result was the same: the distribution of KS arrows for the reshuffled sequences is typical of a pair with no leader-follower interactions, i.e., with low arrow values, centered around but not equal to zero (see Fig. 3). Actually if the pair of sequences was coming from two independent random sequences, the distribution of arrows for the reshufflings is identical to the distribution of KS arrows for an ensemble of independent pairs of sequences (Fig. 3 A); while for the case of correlated sequences both distributions differ (Fig. 3 B). This result opens the opportunity to discern whether a KS arrow $A_{KS}$ is significant, as one can compute the $p$-value of the KS arrow found for the real pair of sequences by calculating the amount of probability that is above $|A_{KS}|$ and below $-|A_{KS}|$ in the distribution of KS arrows coming from the reshufflings. This is the probability that a value higher than $|A_{KS}|$ or lower than $-|A_{KS}|$ comes from a pair of random sequences.

## 2.3 A case study: reef manta rays (Mobula alfredi) at a cleaning station

We applied the described leader-follower analysis methodology to presence data collected for acoustically-tagged reef manta rays (Mobula alfredi) at a cleaning station at a remote coral reef in Seychelles. The resulting social network and reported leader-followers behavior were then examined relative to the sex and size of tagged manta rays.

### 2.3.1 Data collection

Presence data for reef manta rays were collected using passive acoustic telemetry, whereby a single acoustic receiver (VR2W; VEMCO) was placed in close proximity to a known cleaning station at D'Arros Island, Seychelles [21]. Twenty-five acoustic tags (n = 4, V12; n = 21, V16-5H) were externally deployed on free-swimming reef manta rays using a using a modified Hawaiian sling between April 2013 and March 2014. Tags anchors were made of either titanium (n = 4) or stainless steel (n = 21), and were positioned towards the posterior dorsal surface of each individual. Prior to tag deployment, a photograph was taken of the unique and consistent spot pattern present on the ventral surface of each manta ray [43, 49] to allow for individuals to be identified in future monitoring surveys. The sex of each individual was determined by the presence (male) or absence (female) of claspers, and size (i.e. wingspan) was visually estimated to the nearest 0.1 m. Individuals were classified into one of the following two size groups based on their estimated size: small (1 female, 9 males), and big (9 females, 6 males).

Detection data from the cleaning station receiver were downloaded every six months, and once a year the battery was replaced and the receiver inspected for damage or clock drift. Preliminary range testing indicated a detection radius of approximately 150m (mean = $(165 \pm 33)$m [27]) at the cleaning station receiver. Upon import of the detection data into a Microsoft Access database, false positive detections were removed through filtration for active tags and any receiver clock-drift time corrections were calculated assuming linear drift [27].

### 2.3.2 Leader-follower network

To construct the leader-follower network, we analyzed the interactions occurring between pairs of tagged manta rays. Interactions were defined as the presence of one individual and a consecutive appearance of the other. For statistical purposes, we considered only the pairs with more than 50 interactions ($n = 12$ individuals). We then assessed value and direction of the KS arrows between the distribution of lag times for each pair, as described above. We used a global reshuffling scheme to compute the p value associated with each KS arrow, and discarded

those for which $p > 0.002$. This reshuffling scheme allowed for the number of detections for each individual to remain fixed, but for the timing of the events to be reshuffled from the pool of all detections. This differs from a local reshuffling scheme, which was also trialed and where only the two sequences involved in the calculation of the arrow are involved in the reshuffling, however, both schemes lead to similar results. In order to assess the correlation of sex and individual size with the topology of the network, we compared two basic quantities: (1) the number of edges present in the network of each type and (2) the average weight of the edges. Comparisons were drawn across 100,000 reshufflings of the sexes and sizes, respectively, but keeping the network structure fixed.

## 3    Results

### 3.1    Case study: reef manta rays at a cleaning station

A total of 41,607 detections of acoustically-tagged reef manta rays were recorded at the cleaning station receiver between April 2013 and January 2016.

### 3.2    Temporal heterogeneities

The manta detection data were not distributed evenly in time, but rather in a quite peculiar fashion. In Fig. 4A we show the distribution of events for mantas with more than 100 records ($n = 18$ individuals). Detections are most commonly recorded around noon, with an evident circadian pattern (Fig. 4 B). The timing between consecutive events of single individuals, hereafter 'interevent times', are distributed following a heavy-tailed distribution, with a tail consistent with a power law of exponent 1.38 (Fig. 4 C). The exponent has been fitted using the 'powerlaw' package available for python [50]. This implies that the temporal appearances of the manta rays at the cleaning station are burst-like, with periods of high activity followed by long times of inactivity and individual absences.

### 3.3    Leader-follower network.

The final network constructed for reef manta rays at the cleaning station consisted of 12 individuals; five females (one small, four big) and seven males (3 small and 4 big). A total of 33 edges, representative of calculated KS arrows, were included in the network to generate a single component (Fig. 5 A). Regarding the sexes, mixed edges are on average stronger than expected, much more so for males following females, although there are many fewer males following females than expected, in contrast to females following males, which are overrepresented. For same sex edges, the results reflect a randomized scenario. That is, while the number of edges present is as expected, their strength is smaller than expected (see Fig. 5 B). As for the sizes, the most salient result is that the average weight of the interactions of small individuals following small individuals is much lower than expected. The other types of edges are only slightly off the values that are expected. So, small individuals following big individuals do so at a stronger frequency than expected and there is one link less than expected. For big reef manta rays following small ones, the strength is slightly weaker than expected and there is only one extra edge. For big individuals following other big conspecifics, the strength of the interaction is slightly higher than expected, and there are only a couple more edges than expected (see Fig. 5 C). In Fig. 6 we show the average appearance rate averaged over all pairs. The appearance rate is computed for one individual as a function of the time since the last presence event of the other individual, but restricting to pairs for which there is a connection in the inferred network. One can see that the appearance rate for followers is bigger than

for leaders up to a certain time lag of around 200 minutes. This enhanced appearance rate of the following individuals after the appearance of the individual that they follow confirms our measurement of leader-follower behaviors and discards that we are not actually detecting avoidance situations.

# 4  Conclusion

Leadership analyses represent a source of behavioral information. For example, the presence or absence of leadership between adult and juvenile individuals provides insight into the learning strategies of a species, which can be biased towards learning from adults (i.e., juveniles following them), or towards learning through trial and error (i.e., displaying independent behavior) [8]. Additionally, leadership analyses can be used to reveal long-term hierarchies in animal groups and aid in characterizing the structure of populations [10].

Here, we introduced a statistical method for the inference of leadership patterns recorded in presence data collected at a single location. The method relies on randomizations of the input data, which provide estimates for the p-values associated with the leader-follower interactions found. We have tested this methodology against intuitive models of leading behavior, and applied it to a real-world scenario involving presence data for reef manta rays at a cleaning station recorded using passive acoustic telemetry. Our method allowed us to construct a directed network from the manta detection data, which represents leader-follower interactions within the tagged population, and examine whether covariates such as sex and size affect the position of the individuals in the network. Female manta rays were found to follow more males than expected, while males followed fewer females, but with a stronger association. With respect to size, smaller mantas followed other small mantas with a weaker interaction strength, whereas the rest of interactions within the network appeared with a similar pattern as that expected from the randomized case. Furthermore, we examined whether there is an enhanced probability of the appearance of a follower after the appearance of a leader at the cleaning station appeared at the location, and confirmed that the associations found within the study are of the a leader-follower nature.

*Comparison with other similar works.* Ref. [34] examines the associations among reef manta rays based on sighting data recorded using photo-identification techniques at five locations in Indonesia for five years. This presents a unique opportunity to compare sampling, network analysis techniques, and findings between two populations of reef manta rays. While the photo-identification data set of Ref. [34] spans a longer temporal range than that of the present study, data collection was restricted to daylight hours only. In contrast, the automated data collection facilitated by the acoustic receivers and tags of this study allowed for presence data to be collected across the entire diel cycle. This allowed for a more detailed and dynamic approach to the examination of leader-follower interactions in reef manta rays to be developed, and made it possible to move beyond traditional associations defined using the "Gambit of the Group" (i.e. assuming that all individuals observed together are associated). Note that this difference is crucial, since in our framework, two individuals, even if observed always together, might not be related by a leader-follower interaction depending on the timing of the presence events.

Nevertheless, it is interesting to compare results between populations and datasets in terms of the influence of sex and size on network position. Perryman *et al.* found that male reef manta rays tend not to associate with other males, and for them avoidance is common, whereas females may be associated significantly with other females. The highest percentage of preferred dyadic associations was given between different sex pairs, however, there was partial sexual segregation. Regarding maturity status, Perryman *et al.* found that juveniles tend to

associate in the short term with other juveniles and mature adults. It is challenging to determine whether the differences in findings between this, and the present study, are the result of natural behavioral differences or are artifacts of differing analysis procedures. Should acoustic data become available for reef manta rays at any of the five Indonesian field sites used by Perryman *et al.*, future works should aim to replicate the leader-follower analyses outlined here and to compare the results with the findings of the photo-identification-based study.

Ref. [32] describes a method for extracting social structure information from data similar to that of the present study, but again, relies on the Gambit of the Group theory. The major drawback of this method (available within the package GMMevents) is that it relies on clustering the observations of assocations into different event windows, as observations occur in bursts. In many situations, these events windows may be very difficult to define, as the inter-event times of the observations appear power-law distributed, without a clear cut for the clusters, and it may be that association data is lost in the process. Ref. [14] builds upon the methodology described in in Ref. [32], and allows for 'leadership' behavior to be inferred. This is achieved by investigating which animal appears first in the association events that the clustering method has identified, however, by doing so, eliminates all of the information about the dynamics contained in the timings of observations inside the event. Our method does not separate times into classes (such as being part of an event), and instead, considers the complete temporal range of the data, making the analysis procedure less prone to ambiguities and information loss while allowing for leadership behaviors to be examined.

*Limitations and future work.* The analytical method proposed in this study allows only for the detection of paired (i.e. dyadic) interactions from presence data collected at a single location, however, cannot currently be used to detect more complex, collective behaviors. The expansion of these analysis to encompass data collected by multiple acoustic receivers placed at different locations (i.e. acoustic arrays; e.g [21]) will allow for leadership networks to be described using a multilayer approach [51],whereby each receiver is considered as a different layer. Such an approach may reveal if leadership patterns are coherent among different locations that may be significant for different ecological reasons (e.g. cleaning, feeding), if followers continue to follow their leader across wider spatial scales, or if interactions are more complex and followers in one location become leaders in another. For now, however, the present methodology can be applied to existing acoustic telemetry data sets around the globe, and may reveal previously unidentified leader-follower behaviors and patterns in marine species across various ecosystem types.

**Author contributions**  JFG, JPR, and VME designed the research. LRP provided acoustic telemetry data. JFG, JPR, KK and VME developed the methods and analysed the results. All authors contributed to the writing of the manuscript.

**Funding information**  JFG acknowledges funding from the University of the Balearic Islands through its postdoc program and the postdoctoral program Vicenç Mut from the Direcció General de Política Universitària i Recerca from the government of the Balearic Islands. KK acknowledges funding from MINECO through the Ramón y Cajal program and through project SPASIMM, FIS2016-80067-P (AEI/FEDER, EU). JFG, JPR and VME acknowledge funding from the Office of Sponsored Research at King Abdullah University of Science and Technology (KAUST) through project CAASE. Field research in Seychelles was approved by, and conducted with the knowledge of, the Seychelles Bureau of Standards, the Seychelles Ministry of Environment, Energy and Climate Change. Fieldwork and data collection was funded by the Save Our Seas Foundation (SOSF) and supported by The Manta Trust and SOSF – D'Arros Research Centre (SOSF-DRC).

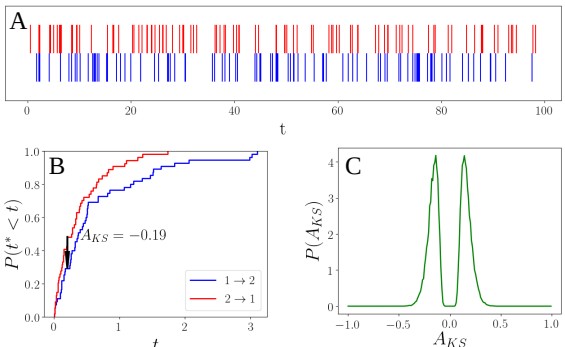

Figure 1: (Color online) Two independent homogeneous Poisson processes of rate $\lambda = 1$. The sequences were generated up to a maximum time of 100 time units. **A** Raw event data. Individual A in blue and Individual B in red. **B** Cumulative distribution of lag times and Kolmogorov-Smirnof arrow, $A_{KS}$, obtained from the sequences shown above. In blue is the cumulative distribution of lag times of Individual A following Individual B, while in red is its conjugate (lag times of B following A). **C** Distribution of Kolmogorov-Smirnof arrows from 105 realizations of pairs generated as in the sequences above. The distribution shows two peaks around 0, signature of no leader-follower relation.

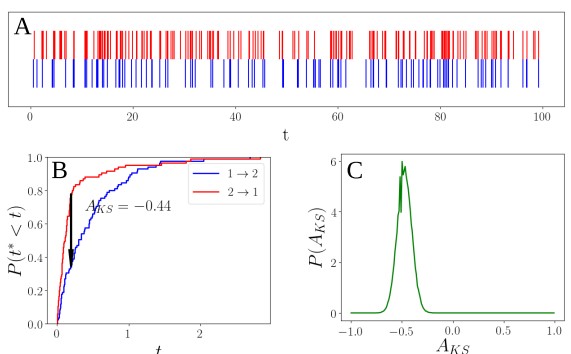

Figure 2: (Color online) Correlated time sequences. Individual A (blue) performs a homogeneous Poisson process of rate $\lambda_1 = 1$, while Individual B (red) follows the correlated non-homogeneous Poisson process described in the text with parameters $\lambda^* = 1, \Delta t = 0.2$ and $\delta = 4$, i.e., it performs a Poisson process of rate $\lambda_2 = 1$ except for 0.2 units of time after an event of Individual A, when it performs a Poisson process of rate $\lambda_2 = 5$. The sequences were generated up to a maximum time of 100 time units. **A** Raw event data. **B** Cumulative distribution of lag times and Kolmogorov-Smirnof arrow for the sequences shown above. **C** Distribution of Kolmogorov-Smirnof arrows from 105 realizations of pairs generated as in the sequences above.

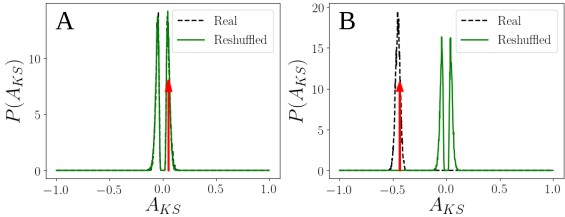

Figure 3: (Color online) Assessing significance of the leader-follower relation. **A** Random uncorrelated sequences with $\lambda_1 = \lambda_2 = 1$ and $t_{\max} = 10^3$. In dashed black lines the distribution of KS arrows for the ensemble of those sequences ($10^4$ independent realizations). In green is the distribution of KS arrows for $10^4$ reshufflings of the pair of sequences that gives rise to the arrow marked in red. **B** Correlated sequences with $\Delta t = 0.2, \delta = 4$ and a maximum time of $10^3$ time units. The KS arrow distribution for these ensemble of sequences is shown in black ($10^4$ independent realizations), while in green is the distribution of arrows coming from $10^4$ reshufflings of the sequence pair that gives rise originally to the value of the arrow signaled at the red line.

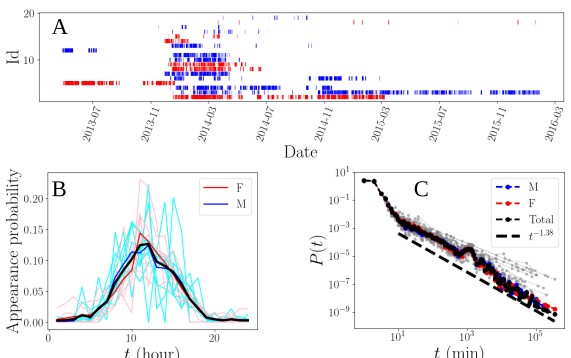

Figure 4: (Color online) Temporal heterogeneities in the data. **A** Raw event data for acoustically-tagged reef manta rays (Mobula alfredi) with more than 100 detection events at a receiver located near a cleaning station in Seychelles. **B** Appearance probability as a function of the hour in the day. The cyan lines correspond to different males, while the pink lines correspond to different females. The red and blue lines are the averages for females and males respectively. The black curve corresponds to the appearance probability of all individuals pooled. We can see that the appearance of reef manta rays at the cleaning station is concentrated around noon. **C** Interevent times distribution, i.e., the distribution of times between consecutive presence events of the same individual. The long tail, compatible with a power law of exponent 1.38 makes clear the burst-like presence of manta rays at the cleaning station. The peak in the distribution is located at a time of one day, which is a reflection of the circadian pattern (i.e. 24 hr) of the manta rays.

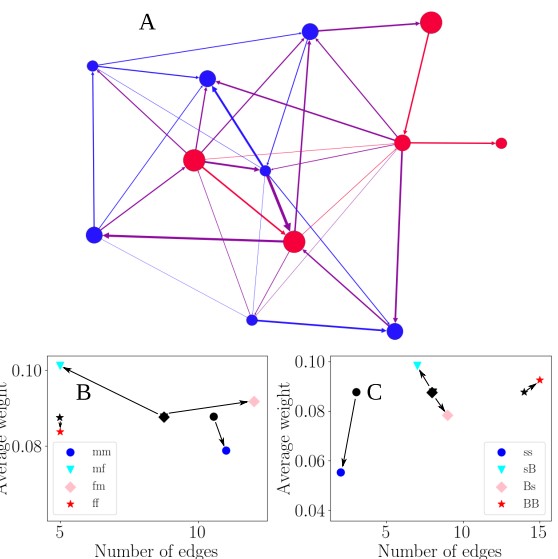

Figure 5: (Color online) Leadership network for acoustically-tagged reef manta rays (Mobula alfredi) detected by a single acoustic receiver placed at a cleaning station at a remote coral reef in Seychelles relative to the sex (Female, pink; Male, blue) and size (node size) of individuals. **A** Leader-follower network of manta rays at a confidence value $p = 0.002$. **B** and **C** Types of edges depending on sex and size. An edge of type xy stands for an individual of type x following an individual of type y, where x and y can be f (female) or m (male) for the sexes, or s (small) and B (big) for the sizes. The black symbols are the expected values from $10^4$ reshuffling of the sexes/sizes of the individuals. The x-axis shows the number of such edges, while the y-axis shows the average strength of the edges. Deviation of the real data in the x direction indicates a difference of the number of edges of that kind found in the real network. Deviation in the y direction signals a difference with the expected strength of the leader-follower relation. The arrows show how the results from the original network differ from the results of randomizations.Results for the randomization of sexes (**B**). Mixed edges are on average stronger than expected, and there are much less males following females than expected, in contrast to females following males, which are overrepresented. For same sex edges the results are similar, the number of edges is approximately the one expected, while the strength is smaller than expected. Results for the randomization of sizes (**C**). The most salient result is that the average weight of the interactions of males following males is much lower than expected. The other types of edges are only slightly off the values that are expected. So small individuals following big ones follow a bit stronger than expected and there is one link less than expected. For big ones following small ones, the strength is slightly weaker than expected and there is only one extra edge. For big ones following big ones the strength is slightly bigger than expected and there are only a couple more edges than expected.

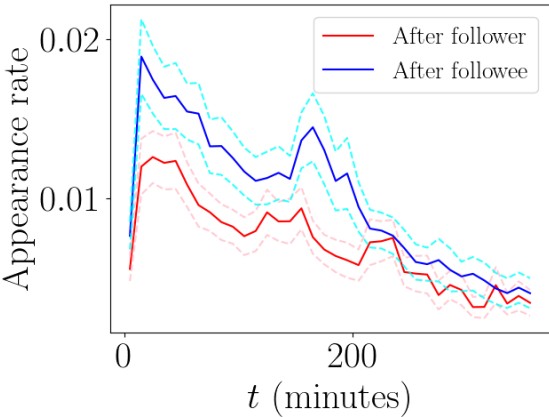

Figure 6: (Color online) Appearance rate for the follower right after the appearance of the leader (blue) or for the leader after the appearance of the follower (red). The rates show that the follower actually has a higher rate of appearing right after an event of the leader. The curves show the average for all the pairs present in the network. The dashed lines show the standard errors.

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
