# Peer review of "Inferring leader-follower behavior from presence data in the marine environment: a case study on reef manta rays"

_SciPost Physics_

## Round 1 · Referee Report · Anonymous (Referee 1) · 2022-8-4

Report

Fernandez-Gracia and coauthors present a general method to create directed networks of follower-leader relationships from the time series of presence data. It is based on the Kolmorogov-Smirnov distance between the interevent time distributions of the agents (nodes). They test their method with synthetic data coming from a Hawkes process and then apply it to a dataset of presence data collected for reef manta rays in the Seychelles. Finally, they draw some conclusions linking the found leader-follower relations in the population of mantas and some metadata, such as the sex or the size.

I would like to start by noting that I have enjoyed reading the article. It is well-written and the length feels appropriate. Moreover, I like interdisciplinary works such as this one, where physics-based methods/concepts (in this case, Hawkes processes, network science) are used to approach problems in other disciplines (in this case, hierarchical relations in an ethological context). I am inclined to recommend the paper for publication, but before doing so I believe that there are some points that still need further clarification and motivation. I present them in the following.

- In the abstract, authors say "more males than expected" and "fewer females than expected". Expected with reference to what? Please specify.

- I wonder whether there is any work or reference the authors can provide where it is motivated the link between physical following (appearance in the cleaning station) and an actual leader-follower roles in some aspect of animal (for the mantas or for other species) culture, such as learning/teaching, mating, etc.

- If I understood correctly, the KS distance is computed between all pairs of mantas, and then those links that present an $|A_{KS}|$ below its value from the reshuffled sequence are ruled out. I would clarify this when the algorithm is introduced, Section 2.1.

- The notation is confusing, since the authors call $t$ both the absolute time (e.g., in the introduction of the models in Sect. 2.2, in Figs. 1A, 2A, 4B, etc.) and relative (interevent) time (e.g., in Figs. 1B, 2B, 4C, etc.). Please solve this by using different symbols.

- For the sake of completeness, please indicate that red and blue in Fig. 4A correspond to female and male (if that is the case).

- I understand how the toy model based on heterogeneous Poisson processes mathematically works, but I find it hard to translate into the physical terms of the manta rays. If Individual A activates and B is to follow her, then the latter will have an increase in her rate during an interval $\Delta t$. This means that during $\Delta t$ the detector should detect B more often once A activates, i.e., A is detected and B has more chances of going in and out of the detector range. Why? If B is following, she should not enter and exit the detector range less frequently? How do the authors justify that this model is a "presence" toy model?

- Regarding the previous point, the authors should specify what they understand by "following", since the detector has a range of 300m of diameter. Is the following close in space and time? Do both manta rays follow similar trajectories in space but with a (more or less) constant time lag, like a kind of marine version of pheromones?

- I miss some more information regarding the longitudinal dimension. The authors differentiate mantas in size (small and big), which I interpret as a proxy for their age. The entire time range they are looking at is close to 3 years. How much, on average, a manta ray live? Can be safely assumed that small/big individuals remain so during the whole observation period?

Requested changes

Please address the points that I mention in the above report.

---

## Round 1 · Referee Report · Anonymous (Referee 2) · 2022-10-28

Report

In this work, Fernandez-Gracia et al. introduce a mathematical tool to infer Leader-Follower (L-F) relationships based on presence data in an ethological context. The kind of data available has recorded the time at which specific individuals (in this case, of manta rays) are present nearby a specified area (here, a cleaning station). The way in which the data is collected and the species involved are tangential to the method developed, so that this analysis could be used elsewhere. The method to infer L-F dynamics depends on a leader (say animal a_i) prompting, in average, that its follower (say animal a_j) is present near the designated area more often than by chance. If that is the case, the average time interval between a_i arriving at a place and a_j following should be smaller than the average time interval between a_j arriving at a place and a_i following. The Authors capture this statistical tendency through a signed Kolmogorov-Smirnov distance (dubbed a KS arrow), which measures differences in the cumulative distribution of time intervals between animal visits. This allows them to associate a directed connection to pairs of animals that would capture the intended relationship. The Authors validate their methods with a toy model in which they manually introduce the desired L-F dynamics—thus they can be sure that their tool captures such relationships indeed. Later, the Authors use their method to build an L-F network from a dataset of manta ray visits to a cleaning station. Their analysis allows them to derive interesting ethological information about the manta ray group under research.

Barring a few comments attached below, I think that the paper is well written and merits publication. Furthermore, I think that the methodology introduced can be of importance to a broad range of researchers that are interested in reconstructing L-F networks (and, more generally, hierarchies within animal communities). This is a non-trivial problem, and treatment of empirical data is plagued with unexpected problems. As such, new methodology in this direction (especially, as it is the case, if based on rigorous statistical tenets) is always of great interest.

I think that the paper would greatly benefit if the following points were addressed:

• It is not straightforward to see why the KS arrow would encode L-F relationships. I can see it after making a few visualizations to help me simplify the concepts involved—which are non-trivial. I think that the paper would improve greatly if there were a figure that would exemplify (perhaps with cartoons, or with toy-model trajectories) how L-F relationships result in larger KS distances. Illustrating this point very clear could make the methodology much more appealing to others who might be interested in using it.

• As far as I understood, the correct way to asses L-F relationships with this tool would not be through a single KS arrow, but through a distribution of KS arrows. The authors exemplify in Fig. 1C how a case with no L-F dynamics results in a distribution of KS arrows that is symmetric around 0, but bimodal and with non-zero modes. This means, as they report, that single instances drawn from that distribution would result in a non-zero KS arrow. They deal with this problem cleverly by comparing KS arrows to distributions obtained through reshuffling the data. This allows them to derive boot-strapped p-values. This approach seems very convincing. However, I wonder if there might be other alternatives to deal with this problem, and whether more information might be extracted from them. Specifically:
◦ Using Jackknifing on the original data should allow us to build KS-arrow distributions. Doing this, asymmetric distributions should stand out promptly. Hence this could be a quick, reliable indicator of whether an L-F relationship exists, and this might require less computations than the reshuffling.
◦ The method relies on the KS distance, which is the largest vertical distance between two cumulative distributions. I wonder whether it could be useful to look at the whole distribution of vertical distances between two given cumulative distributions. The bimodal plot in Fig. 1C (where no L-F dynamics are present) results from a cumulative distribution having, just by chance, a largest cumulative distribution than another at some value of t. Given the underlying stochasticity in this, we could expect that these two cumulative distributions might cross each other several times. On the other hand, in an actual L-F relationship, we would expect that the cumulative distribution for the leader is always larger than the cumulative distribution for the follower for any value of t. In other words, I pose the following question: Could distributions of vertical distances across cumulative distributions result in bimodal plots when no L-F relationship exist, while presenting a clear, skewed, unimodal plot when an actual L-F is present? If this were the case, the Authors could derive L-F relationships robustly without needing to reshuffle the data.

• Regarding the network reconstructed with the actual manta ray data: Several times in the abstract and throughout the paper, the Authors state that the network resulted in more or less connections than expected, or that connections are weaker or stronger than expected. Another Referee has requested some clarification with respect to this point, and I agree that a better explanation is needed. After reading the paper, I assume that this is addressed in the caption of Fig. 5B-C. I think that this is a piece of information important enough that it should be included in the main text. For example, the whole paragraph of section 3.3 relies on knowing the baseline that serves for comparison.

• To the best of my understanding, I think that the methodology would not be able to distinguish between a case with no L-F relationship and a case in which two animals take turns as leaders and followers. Is this correct? I think that the method and results are relevant and robust notwithstanding this—in other words, this is not an issue for the paper. Inferring L-F dynamics is difficult enough, and other methods (as cited in the manuscript) would incur in the same or even worst shortages. But I think that the Authors should mention this case if it is relevant (or discuss how their method solve the problem, if it does).

• There are a few style issues that I think should be revised:
◦ In the Abstract, lines 22-23: I think that that sentence (marked iii) is not very clear.
◦ In the Introduction, lines 66-68: This is a bit difficult to follow.
◦ In the explanation of the toy model (first paragraph in section 2.2), I had a few difficulties understanding what has been done. I feel that the model is simple enough that it could be explained in easier terms.
◦ All three subsections in the Results refer actually to the same case. Does it really pay off to separate the text like that? I think that all could be put under one single header. Also, the single paragraph comprising section 3.3 could result in a better read if it would be split somehow. There is a lot of redundancy (same words; following, followers, individuals, etc.) that at points it becomes a bit confusing. Perhaps separating into paragraphs could help.

Requested changes

I think that the paper is quite correct as it is, but I also think that it would benefit from addressing some of the issues pointed out in my review. I would leave it to the criterion of the Authors whether to address those issues or not.

---

## Editorial Decision

awaiting_resubmission